# Spontaneous ordering of identical materials into a triboelectric series

Juan Carlos Sobarzo[1 ✉], Felix Pertl[1], Daniel M. Balazs[1], Tommaso Costanzo[1], Markus Sauer[2], Annette Foelske[2], Markus Ostermann[3], Christian M. Pichler[3], Yongkang Wang[4], Yuki Nagata[4], Mischa Bonn[4] & Scott Waitukaitis[1 ✉]

When two insulating, neutral materials are contacted and separated, they exchange electrical charge[1]. Experiments have long suggested that this 'contact electrification' is transitive, with different materials ordering into 'triboelectric series' based on the sign of charge acquired[2]. At the same time, the effect is plagued by unpredictability, preventing consensus on the mechanism and casting doubt on the rhyme and reason that series imply[3]. Here we expose an unanticipated connection between the unpredictability and order in contact electrification: nominally identical materials initially exchange charge randomly and intransitively, but—over repeated experiments—order into triboelectric series. We find that this evolution is driven by the act of contact itself—samples with more contacts in their history charge negatively to ones with fewer contacts. Capturing this 'contact bias' in a minimal model, we recreate both the initial randomness and ultimate order in numerical simulations and use it experimentally to force the appearance of a triboelectric series of our choosing. With a set of surface-sensitive techniques to search for the underlying alterations contact creates, we only find evidence of nanoscale morphological changes, pointing to a mechanism strongly coupled with mechanics. Our results highlight the centrality of contact history in contact electrification and suggest that focusing on the unpredictability that has long plagued the effect may hold the key to understanding it.

Contact electrification, also known as 'tribocharging' or 'triboelectrification', defies our understanding. In principle, it seems simple: take two neutral insulators, touch and separate them and they will exchange electrical charge[1–3]. Often identified with 'static electricity' and demonstrations of balloons rubbed on hair, contact electrification is nevertheless essential in wide-ranging corners of nature, from the electrification of thunderclouds[4], to the pollen that sticks to bumblebees[5], to the accretion of dust into protoplanets[6]. Yet, as any contact electrification article reiterates, the most fundamental aspects of the effect, that is, the charge carrier(s) and the cause(s) for their exchange, remain debated. Among the most salient observations associated with contact electrification is the 'tendency' of different materials to order into triboelectric series, that is, transitive lists based on the sign with which materials charge[2,7–10]. For example, in the first such list, created by J. C. Wilcke in 1757, glass charged positive to paper and paper charged positive to sulfur, ergo glass charged positive to sulfur[2]. The notion of triboelectric series has prompted suggestions that contact electrification might be dominated by a single underlying parameter—for the sake of a name, call it $\varphi$. The slate of candidate mechanisms associated with $\varphi$ is numerous, including: electronic properties[11,12], acidity/basicity[13–15], zeta potential[16,17], hydrophilicity/hydrophobicity[18–23], flexoelectricity[24] and mechanochemistry[25–31], to name a few. However, owing to the lack of consensus on the most basic aspects of contact electrification, there is no agreement that one,

if any of these, is correct. Moreover, many experiments cast doubt on the validity of triboelectric series—and the hope for any rational explanation of contact electrification—all together[1,3]. Series compared among different laboratories are frequently inconsistent[1,3,32]. Two materials can initially exchange charge one way (A positive to B), only later to exhibit polarity reversal (B positive to A)[33,34]. Experiments have occasionally pointed to the existence of 'triboelectric cycles', that is, series that loop back onto themselves[25,35,36]. More perplexingly still, even 'identical' materials exchange charge when contacted[37–39], with one leading model for this effect relying on randomness and unpredictability[40].

On the basis of the preceding discussion, our investigation begins with the following question: do samples of identical materials order into triboelectric series? We pursue this question with the system shown in Fig. 1. We prepare identical samples of polydimethylsiloxane (PDMS) and label them with letters A–H (Fig. 1a). We work with PDMS because of its low Young's modulus ($E = 4.3 \pm 0.2$ MPa; see Supplementary Information and Extended Data Fig. 1) and extreme smoothness (nominal roughness of pristine, that is, newly fabricated and uncontacted, samples $R_q \approx 7$ Å), which help to make contacts as 'conformal' as possible. To measure charge exchange for a pair of samples, we use the setup shown in Fig. 1b. We mount each sample on a polytetrafluoroethylene (PTFE) rod. One rod is inside a Faraday cup that is connected to an electrometer, which allows us to measure charge (see Methods).

[1]Institute of Science and Technology Austria, Klosterneuburg, Austria. [2]Analytical Instrumentation Center, TU Wien, Vienna, Austria. [3]Centre for Electrochemistry and Surface Technology, Wiener Neustadt, Austria. [4]Molecular Spectroscopy Department, Max Planck Institute for Polymer Research, Mainz, Germany. ✉e-mail: jsobarzo@ist.ac.at; scott.waitukaitis@ist.ac.at

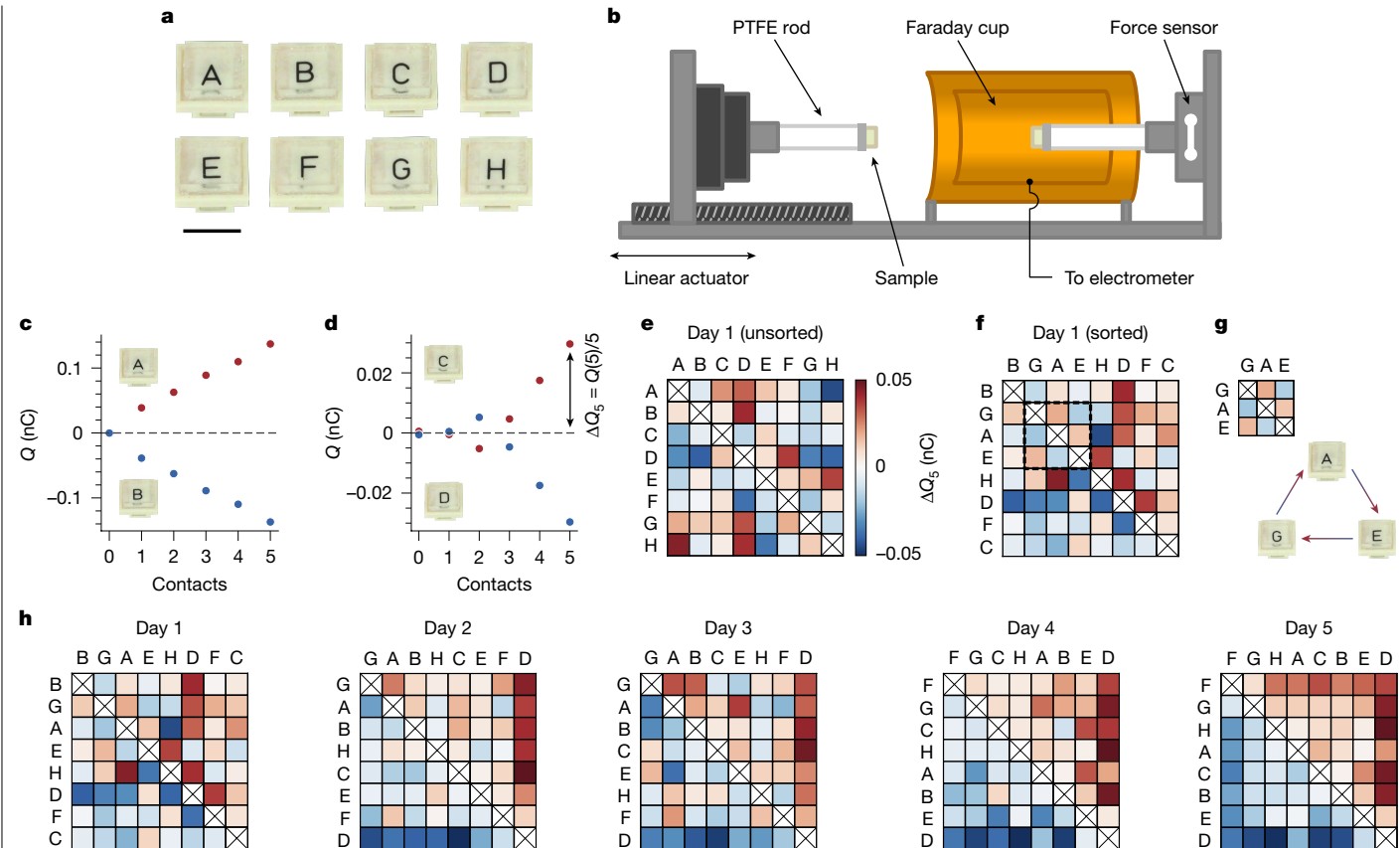

**Fig. 1 | Identical materials spontaneously order into a triboelectric series.**
**a**, We prepare identical samples of PDMS by curing a single parent slab and then cutting eight smaller (1 cm × 1 cm × 0.3 cm) pieces, labelled A–H. Scale bar, 1 cm. **b**, We mount sample pairs on PTFE rods and use a linear actuator to push them together. Inline force feedback allows us to reach a set pressure ($P \approx 45$ kPa, approximately 1% macroscopic strain). We measure charge exchange with an electrometer connected to a Faraday cup enclosing one sample. Before all measurements, samples are discharged to <0.5 pC (see Methods). The chamber/sample storage area is held at 30 ± 2% RH and 22 ± 1 °C. **c**, Example of charge versus contacts with two pristine samples, one charging persistently positively and the other negatively. **d**, Charge exchange between two pristine samples that exhibits an unpredictable sign change. To average over such variability,

we define the average exchange after five contacts as $\Delta Q_5$, which is always measured after both samples have been completely discharged. **e**, To test for a triboelectric series, we measure $\Delta Q_5$ for all pair combinations with a staggered contact sequence (see Methods), creating a matrix in which the colours indicate charge given to the column sample. **f**, The inability to sort the matrix (see Methods) in **e** such that the upper-right (lower-left) corner is purely positive (negative) indicates that the samples charge randomly, that is, do not form a series. **g**, As highlighted in **f**, the defects in a series are indicative of cycles; here A charges positively to G, E charges positively to A, but E charges negatively to G. **h**, Repeating tests over several days with the same set of samples, randomness gives way to order and, by the fifth day, the samples form a perfect triboelectric series.

The other is outside on a linear actuator that enables us to push the samples together. With an inline force sensor, we reach a set pressure ($P \approx 45$ kPa, or approximately 1% macroscopic strain) in every contact. Before measuring charge exchange, samples are always fully discharged (<0.5 pC residual charge; see Methods). Representative charge-exchange data for consecutive contacts with two sample pairs are shown in Fig. 1c,d. In the first, one sample charges consistently positively and the other consistently negatively (Fig. 1c). In the second (Fig. 1d), the charging exhibits a polarity reversal—an observation seemingly indicative of the unpredictability of contact electrification but—as will become clear later—an important clue. To account for such variability, we define $\Delta Q_5$ as the average charge exchange over five contacts, always starting with fully discharged samples (see Methods).

We test whether samples order into a series by measuring $\Delta Q_5$ for all pair combinations (28 in total) in a staggered sequence (see Methods) and constructing a charge-exchange matrix, for which the colour represents the charge acquired by the column sample. Figure 1e shows such a matrix for a first attempt with pristine samples. The unsorted matrix, that is, with letters arranged alphabetically, is almost completely random, but this is to be expected. The likelihood of any order corresponding to our alphabetical naming is very small: $1/2^{28}$ if samples

charge completely random, 1/8! if they form an unknown series. When we sort the samples according to the number of positive and negative outcomes (Fig. 1f; see Methods), they still exhibit highly random charging—the columns and rows cannot be arranged such that all entries above (below) the diagonal are positive (negative), as required for a series. On closer inspection, the imperfections are indicative of the triboelectric cycles mentioned previously. For instance, as highlighted in Fig. 1g, sample A charges positive to G, E charges positive to A and yet E charges negative to G. Such paradoxes are not because of remnant charge (as $\Delta Q_5$ is always measured after complete discharge) nor to experimental uncertainty (as measurement error is much smaller than any $\Delta Q_5$ value).

Once again, these data seem to highlight the unpredictability of contact electrification and in particular the role of randomness with 'identical' materials. An optimistic spin on the chaos is that it could offer a starting point from which one might alter parameters one at a time until order appears. To our surprise, however, we need not intentionally alter anything. In Fig. 1h, we show the resulting sorted charging matrices obtained by simply repeating the measurements of Fig. 1e with the same samples on consecutive days. The initial randomness progressively dissipates, giving way to a perfect series on the fifth

attempt. This spontaneous ordering occurs every time we start with a pristine ensemble. Typically, we observe a perfect series around the fifth attempt, although it has required as many as ten and as few as two. Once a series forms, it is relatively stable, but cycles or letter swaps (especially in the centre) may pop in and out.

What causes this evolution? One possibility is that some parameter changes with time, altering the charging behaviour of samples with it. For instance, PDMS is known to exhibit ageing, in which, over a timescale of weeks to months, the mechanical[41] and electrical properties[42] can drift. More generally, contact electrification has been shown to be extremely sensitive to environmental conditions[23], for example, relative humidity (RH) history, which might also be considered. We can exclude such time-based evolution for the following reasons. First, pristine samples begin by charging randomly and evolve into a series independently of when we test them (for example, a day, a week, a month) after their fabrication. Second, although large variations in humidity history (>90% RH) do alter the observed charging behaviour (see Supplementary Information and Extended Data Fig. 2), we store samples and conduct experiments in tightly regulated conditions with little variability (see Methods).

Rather than time, we find that the evolution is a result of the act of contact. Figure 2a shows measurements of $\Delta Q_5$ for 24 pairs of pristine samples. These charge randomly around zero, with a spread of about 0.007 nC. To isolate contact as the culprit, we compare this baseline to experiments with a new batch of pristine samples, half of which we expose to 100 previous contacts (and, as always, discharge). When we measure $\Delta Q_5$ for the 'contact-biased' samples pressed against the uncontacted ones (Fig. 2b), we see a substantial effect. The contacted samples always charge negatively. The 100 contacts used to cause this are well within the total number required for a series to emerge (about 5 trials × 7 samples × 5 contacts = 175 contacts). We investigate this further using a trio of samples, as shown in Fig. 2c: an 'advancing' sample (green A), a 'lagging' sample (ivory A) and an 'extra' sample (ivory X). Starting from the pristine state, we first measure $\Delta Q_5$ between the advancing and lagging samples. We then bias the advancing sample with 20 contacts against the extra sample. After discharging, we measure $\Delta Q_5$ again between advancing and lagging samples. By repeating this process, we generate a growing difference in the contact history between the advancing and lagging samples. This translates into the advancing samples charging increasingly negative, with different rates and plateaus for different trios (Fig. 2d). As we show in the Supplementary Information and Extended Data Fig. 3, the bias created by contact is long-lasting. The act of contact—by definition required for contact electrification to occur—can alter the underlying parameters(s) that drive contact electrification. In other words, our materials 'remember' their contact history.

This reality suggests a hypothesis to explain the spontaneous ordering into a series. First, we suppose that every sample, $i$, has some 'effective potential'—for the sake of a name, call it $\varphi_i$. In line with the randomness-based model for identical materials[40], their initial values come from some parent distribution, $\varphi_i^0 \approx \mathcal{N}(\varphi^0, \sigma_0)$, which could be, for example, Gaussian[22,40,43]. When two samples, $i$ and $j$, make contact, the charge they exchange is related to the difference in their potentials, that is,

$$\Delta Q_{i \to j} \propto \varphi_i - \varphi_j. \tag{1}$$

In this picture, the ordering of the triboelectric series at any instant should correspond to the ordering of the different potentials. However, it is clear from Fig. 2d that contact causes changes to the potentials, that is, $d\varphi_i/dn_i = f(n_i, \varphi_i, \ldots)$, in which $n_i$ is the number of contacts sample $i$ has experienced. The critical idea is now this: if contact causes the potentials to ultimately separate, then any ensemble will eventually result in a triboelectric series. Moreover, if the ordering of the final

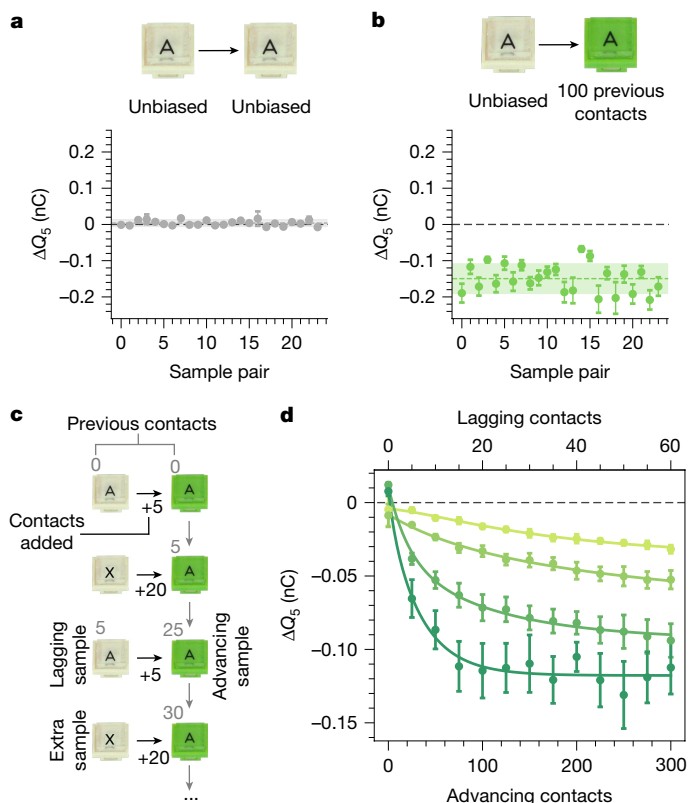

**Fig. 2 | Contact drives evolution. a**, Towards uncovering what causes the series evolution, we first performed baseline measurements between 24 pairs of pristine samples. These charge randomly about zero with a standard deviation of about 0.007 nC. **b**, Motivated by the possibility that the act of contact itself could drive the series evolution, we prepare 48 new samples and expose half to 100 prior contacts. Contacting these against unbiased ones and measuring $\Delta Q_5$ reveals a marked effect, with the previously contacted samples always charging negative. **c**, We investigate how this 'contact bias' evolves using trios of samples: an 'advancing' sample (green A), a 'lagging' sample (ivory A) and an 'extra' sample (ivory X). We first measure $\Delta Q_5$ between the advancing and lagging samples starting from the pristine state. We then subject the advancing sample to 20 contacts with the extra sample. After discharging, we measure $\Delta Q_5$ again between the advancing and lagging samples. Repeating this process develops a growing contact bias between the lagging and advancing samples. **d**, Charge exchange versus advancing (bottom) and lagging (top) contacts for different trios of samples. All advancing samples charge more negatively as their contact bias increases, although each one at a different rate and with a different plateau. As explained in the main text, we assume that the charge exchange in this evolution is caused by a difference in a generalized potential, $\varphi$, which obeys equation (2). Solid curves are fits to the model.

potentials, $\varphi_i^\infty$, is not the same as $\varphi_i^0$, it must be the case that, during their evolution, they 'crossed', resulting in what we (incorrectly) perceive as 'cycles'.

We can use this hypothesis to develop a numerical model that reproduces our experimental data. Figure 2d implies an evolution consistent with the following differential equation,

$$\frac{d\varphi_i}{dn_i} = -\alpha_i(\varphi_i - \varphi_i^\infty), \tag{2}$$

in which $\alpha_i$ are growth rates and $\varphi_i^\infty$ are the values of the potentials at infinite contacts.

We use this equation to fit our data in Fig. 2d, which allows us to extract the ranges of the parameters $\alpha_i$, $\varphi_i^0$ and $\varphi_i^\infty$. We use this information to draw new values of $\varphi_i^0$, $\varphi_i^\infty$ and $\alpha_i$ and define eight 'virtual'

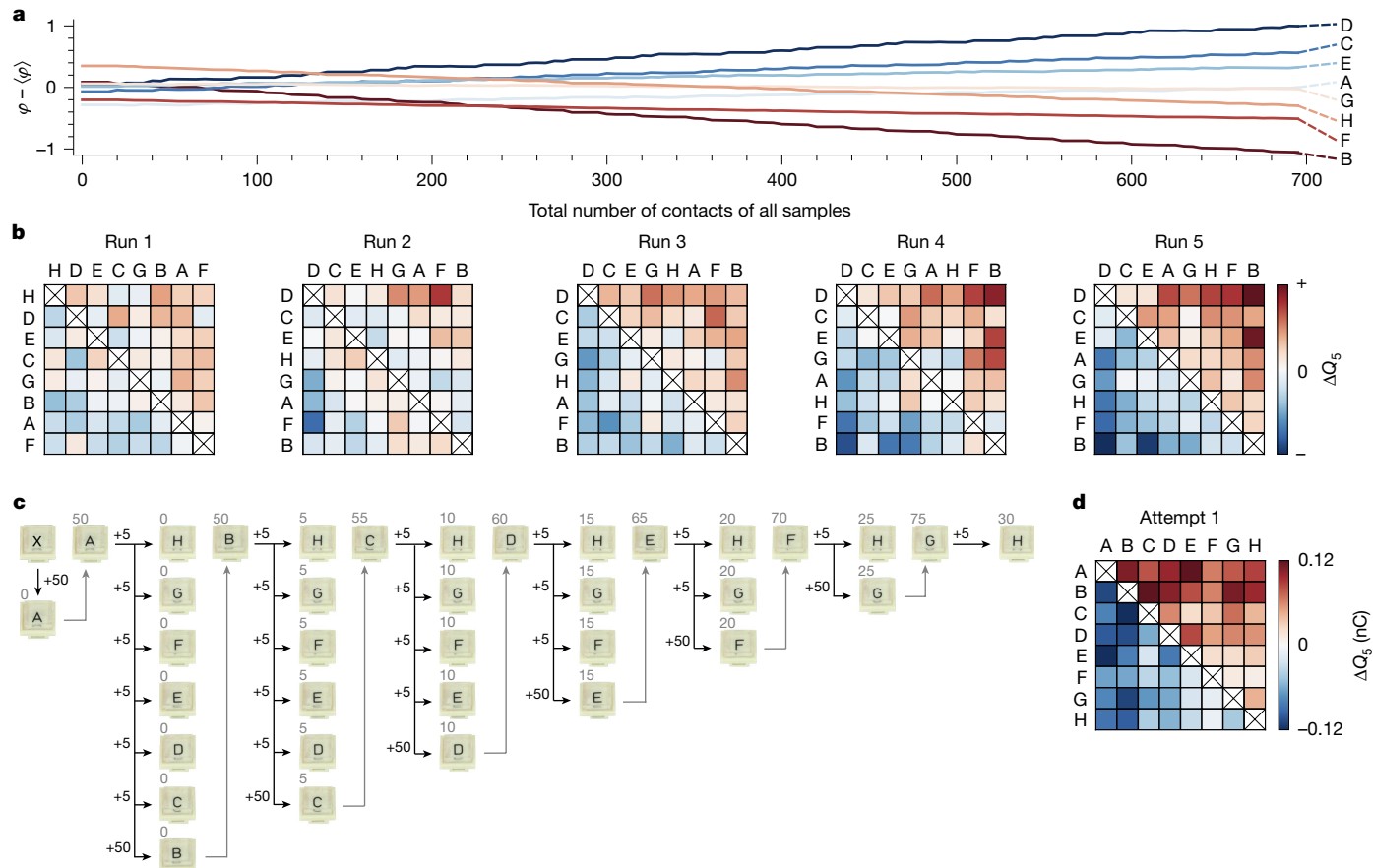

**Fig. 3 | Validating the contact-evolution model. a**, Using equations (1) and (2), we perform numerical simulations to reproduce the series observed in Fig. 1. We initiate eight virtual samples with parameters $\alpha_i$, $\varphi_i^0$ and $\varphi_i^\infty$, drawn from the range of fit values to the data in Fig. 2d. Performing the same staggered contact sequence as in Fig. 1e, we evolve and record all $\varphi_i(n_i)$. We plot the total number of contacts, $N = \sum n_i$, on the $x$ axis and remove ensemble drift on the $y$ axis by subtracting $\langle\varphi\rangle$. Individual $\varphi_i$ swap positions during the evolution but stably separate after a sufficient number of contacts. **b**, Corresponding charging matrices from **a**, for which we observe samples evolve into a series such as in the experiments. The apparent disorder in imperfect matrices, that is, the appearance of 'cycles', is because of the $\varphi$-crossings between samples as

contact causes them to evolve. **c**, We develop an algorithm to force experiments to produce the 'appearance' of a series of our choosing, in this case {A, B, C, D, E, F, G, H}. First, sample A is subjected to 50 contacts with an extra sample X, then five contacts with H, G, F, E, D, C to measure $\Delta Q_5$, and finally 50 contacts with sample B (measuring $\Delta Q_5$ in the first five). Next, sample B has five contacts with H, G, F, E, D and then 50 with C and so on. **d**, As intended, this creates the appearance of an alphabetical series on the first attempt. As we show in Extended Data Fig. 5, this is indeed only an appearance, as reattempting the series with a staggered contact sequence (the one used for Fig. 1e) leads to a different and unpredictable result.

samples. We perform 'contacts' between these samples in a staggered sequence, just as in the experiments (see Methods), calculating the charge exchange through equation (1) and letting the $\varphi_i$ evolve according to equation (2). Figure 3a shows the evolution of the $\varphi_i$ versus the total number of contacts of all samples for a simulation instance. When a sample is being contacted, its potential evolves, and otherwise it is stationary. Owing to the fact that the samples evolve differently, the potentials cross each other. As foreseen in the previous paragraph, triboelectric cycles are manifestations of these crossings—they are not 'real' but rather consequences of our inability to perform contact without changing the system. This is the clue hidden in Fig. 1d, in which we witness a crossing in the act. As the samples continue to evolve, the individual $\varphi_i$ slowly separate, until they are sufficiently resolved and no more crossings occur—the series is established. The corresponding matrices can be seen in Fig. 3b, in which—after five runs—a perfect series is attained. Just as in the experiments, the speed of this evolution depends on the values picked for the model, with slower or faster evolutions possible in different simulation instances.

The proposed model not only allows us to explain previous observations but also predict new ones. To illustrate this, we design an experimental 'contact algorithm' to create the appearance of the series of

our choosing (Fig. 3c). We again fabricate eight pristine samples A–H, plus an extra sample X. As a first step in our algorithm, we use the extra sample to bias A with 50 contacts. We then measure $\Delta Q_5$ for A against the rest of the samples from H to C. With sample B, we perform 50 contacts against A, which simultaneously allows us to bias sample B while measuring its $\Delta Q_5$. Next, we measure $\Delta Q_5$ for B against all other samples except C, with which we perform 50 contacts while measuring $\Delta Q_5$, and so on. As we remarked earlier, the probability that eight pristine samples order into an alphabetical series is very small (1/8! at best). Figure 3d shows that, by understanding and wielding the role of contact history, we create this motif on the first attempt. In the Supplementary Information and Extended Data Fig. 4, we illustrate that we can just as well force the appearance of a cycle by manipulating contact history. In either case, the word 'appearance' is key, as we are enforcing outcomes by always ensuring that one sample has substantially more contacts than the other when $\Delta Q_5$ is measured. As we show in the Supplementary Information and Extended Data Fig. 5, reusing the samples of Fig. 3d for a second try at a series without manipulating the contact order (that is, with the same staggered contact sequence as Fig. 1e) leads to an entirely different and (predictably) unpredictable result, including 'cycles'.

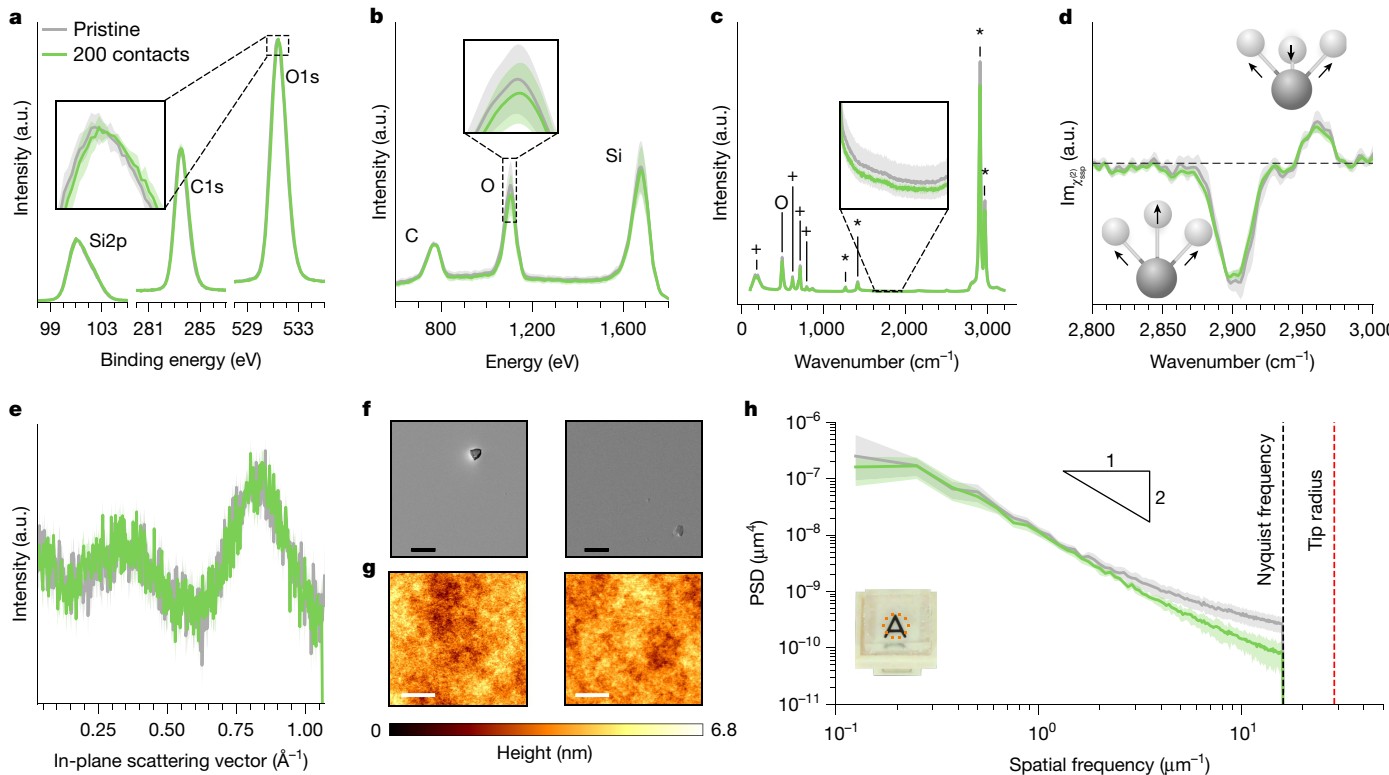

**Fig. 4 | What does contact change? a**, To probe atomic differences in the uppermost approximately 10 nm, we use HR-XPS and measure the Si2p, C1s and O1s peaks. Previous work[44] reported a subtle (about 300 meV) shift in the O1s peak of PDMS after contact electrification with polyvinyl chloride (PVC), but averaging several measurements shows that such shifts are within noise in our experiments (that is, shaded error band in the inset). **b**, Focusing on elemental differences in the outermost atomic layers with LEIS again shows no measurable differences in the C, O or Si concentrations. **c**, To probe for molecular differences, we record Raman spectra at several locations and plot the mean (line) and standard deviation (error band). The peaks marked as +, O and * correspond to Si−C, Si−O and C−H, respectively. We observe no notable differences in any of the peaks, nor can we reproduce differences found previously[30] between 1,600 and 1,950 cm⁻¹ and attributed to COOH groups (inset). **d**, To probe the outermost molecular layer, we use HD-SFG to measure the symmetric/asymmetric C−H

stretching modes (2,900 cm⁻¹ and 2,960 cm⁻¹, respectively). Within the error bands, the pristine/contacted samples are again indistinguishable. **e**, GIXS data for pristine and contacted samples, which probes subnanometre interatomic structure, also renders pristine/contacted samples as indistinguishable. **f**, Using SEM to image each surface, we find no visible changes in the surface integrity; regions with (rare) specks are shown intentionally to aid visualization. Scale bars, 20 μm. **g**, With AFM to characterize surface roughness, we do not find any visible differences. Scale bars, 2 μm. **h**, However, we do detect differences in the PSD of the roughness, for which contacted samples are smoother at higher spatial frequencies than uncontacted ones. The error bands represent scatter from about ten measurements on different regions for the same sample in the pristine/contacted states, indicating that this result is statistically significant. a.u., arbitrary units.

Clearly, the act of contact fundamentally alters sample surfaces in a way that affects the contact electrification mechanism. These alterations could be elemental (for example, changing the atomic composition), molecular (affecting bonds) or physical (for example, changing structure/morphology). In Fig. 4, we present a set of surface-sensitive tests to search for these changes. At the atomic scale, experiments for contact electrification between different materials have shown that contact leads to changes in elemental composition[26,33,44]. Figure 4a shows high-resolution X-ray photoelectron spectroscopy (HR-XPS) data for pristine versus 200-contacted samples, which reveal no statistically significant differences in the silicon, carbon or oxygen content. Increasing surface specificity with low-energy ion scattering (LEIS, which probes the first few atomic layers compared with about 10 nm in XPS) also shows no alterations (Fig. 4b). Some studies have reported molecular modifications after different-material contact electrification[26,30]. We use Raman (to probe about 1 μm into the bulk) and heterodyne-detected sum-frequency generation (HD-SFG) spectroscopy (which probes the topmost roughly 1 nm) and, in both cases, cannot distinguish between pristine/contacted samples (Fig. 4c,d). To test for physical changes, we put three techniques to work: grazing-incidence X-ray scattering (GIXS), scanning electron microscopy (SEM) and atomic force microscopy (AFM). In-plane signals from the GIXS measurements show that

there are no detectable differences in the subnanometre, interatomic arrangements near the surface (Fig. 4e). At first glance, the SEM and AFM images (Fig. 4f,g) seem to indicate no obvious morphological differences on the surface. However, calculating the power spectral density (PSD) of the AFM data (Fig. 4h) reveals a surprising feature−contacted surfaces are smoother at higher spatial frequencies. Performing many scans at several locations on the same sample before/after contact, we find that these differences are statistically significant. In the Supplementary Information and Extended Data Figs. 6−8, we show that increasing from 200 contacts to several thousand creates no changes in the LEIS, Raman and HD-SFG data but causes even more smoothening of the high-frequency tails in the roughness PSD. In Extended Data Fig. 9, we contact pristine samples against intentionally roughened ones, in which again the smooth ones charge negatively. These observations strongly suggest that the high-frequency smoothening presented in Fig. 4h is indeed the cause of the contact bias in Fig. 2 and, consequently, the driver of the spontaneous ordering in our system.

As we discuss thoroughly in the Supplementary Information, a long history of observations casts suspicion on mechanical history and surface morphology, yet have not cleanly established that every charge-exchanging contact probably entangles the two. Most notably, recent experiments have shown the tendency that 'smooth polymers

charge negatively' is widespread[29]. Taking this at face value, the only conceptual leap required for the spontaneous ordering of series to be widespread is the occurrence of morphological alterations during contact, which is a well-established principle in the tribological framework of Bowden and Tabor[45]. These facts lead us to conjecture frequent occurrence in other polymers, although further experiments are required to know for certain. For other insulators beyond polymers (for example, insulating oxides), we do not extend our conjecture.

We conclude that the unpredictability in contact electrification may not be so hopeless after all. By carefully paying attention to contact history, we can not only explain the unpredictability in our system but even tame it. Considering the effect of contact history, the notion of triboelectric series may be a useful heuristic, but not much more. If the effect is widespread, chasing an immutable ordering is comparable with chasing a mirage. Last, by focusing on contact electrification with identical materials, and particular sources of unpredictability therein, we distil minimal ingredients relevant to its mechanism(s). With a phenomenon that is (1) so unpredictable and (2) studied by distinct scientific disciplines with different languages and conceptual biases, such purified information is valuable. In our system, it allows us to conclude that the cause of charge transfer must be intimately connected to nanoscale contact mechanics. In light of this, two of the mechanisms from the introduction deserve special attention: mechanochemistry and flexoelectricity. The former proposes that mechanical strain—which becomes exceptionally large during deformation of nanoscale asperities—can be sufficient to cause heterolytic bond cleavage and consequent liberation of charged species for transfer[25,29,30]. The latter couples electrical polarization to mechanical strain gradients—which also become large in nanoscale deformation—and hence can be expected to produce large electric fields during contact[46]. Our data are insufficient to validate/invalidate either of these hypotheses directly but compel us towards enticing speculation: both may be at play. Our work therefore calls for more careful consideration of these mechanisms, potentially in cooperation, with a special focus on the inherently tribological nature of contact electrification.

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

## Methods

### Sample fabrication

We fabricated PDMS samples by mixing Sylgard 184 elastomer base and curing agent in a 10:1 ratio. After thoroughly blending with a centrifugal, bubble-free mixer (Hauschild SpeedMixer DAC 150.1, 2 min at 2,000 rpm), we poured the uncured liquid into a cylindrical dish to a fill height of 3 mm. We degassed this for 30 min and cured at 80 °C for 24 h. We removed the cured slab from the dish and cut it into square samples of 1 cm side lengths, using a custom-fabricated stencil to precisely guide the blade of a clean, fine razor. We stuck each sample to a 3D-printed holder using more PDMS and further curing at 80 °C for 24 h. For all experiments except those described in the Supplementary Information concerning intentionally modified surface roughness, we used the 'air-facing' side of samples for contact.

### Charge measurement

We measured charge versus time using a Keithley 6514 electrometer connected to the Faraday cup. The charge at the time of each contact (as reported by the force sensor reaching its set point) corresponded to the total charge of both samples, whereas the charge at maximum separation (as reported by a homing device on the linear actuator) corresponded to the charge of the sample inside the Faraday cup. Subtracting charge between these points yielded the charge of the sample outside the Faraday cup. We removed drift by recording the electrometer signal before, during and after the contacts and using interpolation to determine the background trend without contacts. Rather than reporting the charge exchange from a single contact, we always performed five contacts and defined

$$\Delta Q_5 = \frac{\sum_{n=1}^{5} Q(n) - Q(n-1)}{5} = \frac{Q(5)}{5}, \quad (3)$$

in which $Q(n)$ is the charge of the sample inside the Faraday cup after contact $n$. We reiterate that every $\Delta Q_5$ measurement started with both samples fully discharged; residual charge from previous contacts played no role. Taking the average value allowed us to smooth over variability as samples evolved with contacts. The number five was chosen because it is much smaller than the approximately 200 contacts required for samples to evolve fully, but also large enough to yield a meaningful average and permit the series to form in a handful of trials rather than tens or dozens of trials (for example, if we carried out one contact at a time).

### Environmental conditions

Charge-exchange experiments were carried out under highly regulated environmental conditions, either in a custom-fabricated environmental chamber or in an ISO 5 (Class 100) clean room. The environmental chamber consisted of a large (approximately 2 m³) acrylic enclosure with hand-axis ports to manipulate samples. The chamber was equipped with a HEPA filter air circulation system (Levoit LV-H132) to remove trace airborne particles. Humidity in the chamber was maintained at 30 ± 2% RH using a feedback-controlled humidifier (ETS Model 5432) and the temperature was held constant by means of the laboratory conditions at 22 ± 1 °C. For the clean-room experiments, the humidity was held constant at 46 ± 1% RH and the temperature at 22 ± 1 °C. For experiments carried out in both the environmental chamber and the clean room, we (1) observed that pristine samples always ordered into triboelectric series and (2) that the contact bias always evolved as in Fig. 2d.

### Discharge

We used custom-built discharge chambers to remove residual charge from samples before every $\Delta Q_5$ measurement. For the experiments in the main lab, this was directly connected to the environmental chamber through an internal access port, such that samples could be moved between the two without leaving the humidity-regulated environment. For the clean-room experiments, a second discharge chamber was mounted on a rolling cart. The discharge chambers housed photoionizers (Hamamatsu L12645, 10 keV) directed towards a fan. Samples sat behind the photoionizer, in the current of air blown by the fan. As the photoionizer ionized air in front of the fan, this was blown onto the sample. The sample discharged as a result of its electric field drawing in ions of the opposite sign. This is the same manner by which a sample sitting on a shelf would discharge, only accelerated. Notably, this process: (1) works for negatively or positively charged surfaces; (2) works for non-uniformly charged surfaces; and (3) is self-limiting until the charge (or, more stringently, surface electric field) is zero. We characterized the effectiveness of this method by measuring the charge of samples post-discharge in the Faraday cup, which showed the maximum residual charge to be <0.5 pC. This is one order of magnitude below the smallest relevant scale of charging we measure (around 5 pC) for a single contact between two pristine samples.

### Standard 'staggered' contact sequence

When probing for a series as in Fig. 1e, we used a contact sequence ('algorithm') for the pair combinations such that (1) samples were not reused in consecutive measurements and (2) no sample ran too far ahead of the others in its total number of contacts. This sequence was: A-B, C-D, E-F, G-H, B-C, D-E, F-G, H-A, B-D, C-E, D-F, E-G, F-H, G-A, H-B, A-C, B-E, C-F, D-G, E-H, F-A, G-B, H-C, A-D, B-F, C-G, D-H, A-E. We validated by running experiments with many pristine ensembles that this sequence did not influence the result obtained, that is, each time we obtained a different series.

### Matrix sorting

To see whether or not samples formed a triboelectric series after measuring all charging-matrix elements, we sorted the matrix rows/columns according to the number of positive outcomes. When samples do form a series, the result of this sorting is unique: one sample will have seven positive outcomes, one six and so on. When samples do not form a series, the ordering is no longer unique. For example, in the data corresponding to 'Day 4' of Fig. 1h, the positive outcomes are as follows: {D:7, E:6, B:4, A:4, H:3, C:3, G:1, F:0}. Hence there is ambiguity as the placement of A versus B (which both have four positive outcomes) and H versus C (which both have three positive outcomes). As far as we can tell, the relative positions of these rows/columns become arbitrary. However, no matter what order we choose, the inability to order such samples into a perfect triboelectric series remains.

### Contact bias

To probe for the initial contact bias (Fig. 2b), we prepared 48 pristine samples with our usual method. We subjected half of these to 100 contacts among each other before any measurement of charge exchange. We then paired these contacted samples with the remaining uncontacted ones and performed experiments to measure $\Delta Q_5$ in the usual way. As always, all samples were fully discharged before every single measurement of $\Delta Q_5$.

### Contact-bias evolution

To observe the evolution of the contact bias (Fig. 2d), we prepared trios of new samples, naming them 'advancing' sample, 'lagging' sample and 'extra' sample. With these, we first measured $\Delta Q_5$ between the advancing and lagging samples. We then biased the advancing sample with 20 contacts against the extra sample. After discharging, we measured $\Delta Q_5$ again between advancing and lagging samples. By repeating this process, we generated a growing difference in the contact history between the advancing and lagging samples (hence the names). We stress that, before each $\Delta Q_5$ measurement, samples were fully discharged.

Therefore the data points in Fig. 2d do not represent accumulated charge on a sample but rather how having a history of contacts increases the derivative of charge transfer with respect to contact number starting from a zero-charge state (that is, increases $\frac{dQ}{dn}\big|_{Q=0}$).

## Numerical model

In our numerical model, we created eight virtual samples defined by their values of $\varphi_i^0$, $\varphi_i^\infty$ and $\alpha_i$, all chosen to be comparable with the range of fit values from the fits to the experimental data in Fig. 2d. Each sample had corresponding variable parameters $\varphi_i$ and $n_i$, with the pristine state corresponding to $n_i = 0$. We simulated the experiments by pairing the samples in the same staggered sequence as described earlier. When samples $i$ and $j$ came into 'contact', we evaluated the parameters $\varphi_i(n_i)$ and $\varphi_j(n_j)$ using equation (2), calculated the charge exchange with equation (1) and increased the number of contacts. Once all pair combinations were done, we attempted an ordering of the series in the matrix representation (Fig. 3b). We repeated the procedure until we achieved a perfect series, which—as in the experiments—typically took five runs but varied depending on the input parameters.

## Series-forcing algorithm

To force the 'appearance' of the alphabetical triboelectric series in Fig. 3d, we prepared nine new samples: A, B, C, D, E, F, G, H and X. We first used the sample X to bias sample A with 50 contacts. Next, we proceeded to measure $\Delta Q_5$ for A-H, A-G, A-F, A-E, A-D and A-C, but for A-B, we performed 50 contacts while measuring $\Delta Q_5$ (that is, we only used the first five contacts for charge measurement and then used 45 more for biasing). We then measured $\Delta Q_5$ for B-H, B-G, B-F, B-E and B-D, and for B-C, we performed 50 contacts while we measured $\Delta Q_5$. We repeated this process as illustrated in Fig. 3c. As we highlight in the Supplementary Information, this algorithm takes advantage of the sequence of the measurements and the contact bias, allowing us to impose apparent control of the series. In principle, this algorithm is not guaranteed to work all of the time, as we cannot know a priori how fast a given sample will evolve. However, Fig. 2d shows that, after 50 contacts, most samples are sufficiently biased to charge negatively to uncontacted ones, and we had immediate success with this number. If we repeat the measurements with the samples of Fig. 3d using the usual 'staggered' contact sequence (same as Fig. 1e), we find a different and unpredictable result (see Supplementary Information and Extended Data Fig. 5), consistent with the fact that, at the end of the algorithm, we do not know where the potentials are.

## XPS measurements

All XPS measurements were carried out on a PHI VersaProbe III spectrometer equipped with a monochromatic Al Kα X-ray source and a hemispherical analyser (acceptance angle ±20°). Pass energies of 140 eV and 27 eV and step widths of 0.5 eV and 0.05 eV were used for survey and detail spectra, respectively. (Excitation energy: 1,486.6 eV; beam energy and spot size: 50 W onto 200 μm; mean electron take-off angle: 45° to sample surface normal; base pressure: $<7 \times 10^{-10}$ mbar; pressure during measurements: $<3 \times 10^{-8}$ mbar). Samples were mounted on Cu tape. Electronic and ionic charge compensation was used for all measurements (automatized as provided by PHI). The binding energy scale and intensity were calibrated by using methods described in ISO 15472, ISO 21270 and ISO 24237. The analysis depth was estimated to be around 7–10 nm. Surface cleaning was carried out using a gas cluster ion source (5/10/20 kV, 20/30/40 nA). We carried out data analysis using CasaXPS and MultiPak software packages, using transmission corrections, Shirley/Tougaard backgrounds[47,48] and customized Wagner sensitivity factors[49]. We performed deconvolution of spectra using a Voigtian line shape (LA(50)). The line of each spectrum in Fig. 4a represents the mean from XPS scans on four different regions and the error band represents the scatter.

## LEIS measurements

LEIS measurements were executed using an ION-TOF Qtac100 (IONTOF) high-sensitivity spectrometer. As primary ions, $^4$He$^+$ at 3 keV were used at an incident angle of 0° and a scattering angle of 145°. A time-of-flight mass filter was used to improve sensitivity levels, resulting in an effective primary ion current of 452–507 pA. Spectra were recorded between 500 and 2,000 eV. The measurement area was $2,000 \times 2,000$ μm squared. For depth profiling, sputtering was executed with an $^{40}$Ar$^+$ sputter gun at 0.5 keV at an incident angle of 60°, applying a sputter current of 101 nA. Sputter steps of 15 s were implemented between measurements. This resulted in a sputter depth of about 0.005–0.030 nm per cycle. The sputter area was $2,500 \times 2,500$ μm squared concentric around the measurement area. A charge-compensation filament was used during measurements to prevent deterioration of the ion scattering. Data evaluation was performed using the SurfaceLab 7.x software to fit the sample measurements. The line of each spectrum in Fig. 4b represents the mean from LEIS scans on five different regions and the error band represents the scatter.

## Raman measurements

All Raman spectroscopy was carried out on an inVia Qontor spectrometer (Renishaw) equipped with a 532-nm laser. The laser was focused on the sample surface using a 100× objective and a power of 10 mW. Each sample was measured at four different positions, one in the centre and three different corners of the sample, with a $3 \times 3$ grid spaced with 2 μm distance between each point to cover a large surface area. We used SynchroScan wide-range scanning mode to collect data from 100 to 3,200 cm$^{-1}$ with 1,947 measurement points. The line of each spectrum in Fig. 4c represents the mean from data taken on the four different regions and the shaded error band represents the scatter (except for the plasma sample in the Supplementary Information and Extended Data Fig. 7).

## HD-SFG measurements

HD-SFG measurements were conducted using a non-collinear beam geometry with a Ti:sapphire regenerative amplifier laser system (Spitfire Ace, Spectra-Physics, centred at 800 nm, 5 mJ pulse energy, approximately 40 fs pulse duration, 1 kHz repetition rate). A detailed description of the setup is provided in ref. 50. The infrared and visible beams were focused onto a 150-nm-thick ZnO layer deposited on a 1-mm-thick CaF$_2$ window, producing a local oscillator signal[51]. Each spectrum presented in the main text was obtained with a 1-min exposure time and averaged over more than five measurements. All measurements were performed under the ssp polarization configuration, in which 'ssp' denotes s-polarized HD-SFG, s-polarized visible and p-polarized infrared beams. The complex spectra of the second-order nonlinear susceptibility ($\chi^{(2)}$) at the ssp polarization configuration were obtained through Fourier transformation of the HD-SFG interferogram and were normalized by that of a z-cut quartz crystal[52]. In our spectral analysis, we used $\chi^{(2)}_{\text{ssp}}$ data without considering the Fresnel factor and local oscillator reflectivity corrections. The line of each spectrum in Fig. 4d represents the mean from data taken on three different regions and the shaded error band represents the scatter.

## GIXS measurements

GIXS data were collected on a Xeuss 3.0 HR laboratory beamline (Xenocs SAS). Copper Kα radiation from a microfocus source was collimated using a 3D multilayer mirror and shaped by scatterless slits. The beam size at the sample position was $0.5 \times 1.0$ mm squared (horizontal × vertical). The samples were aligned in the beam path and tilted to an incident angle of 0.1° (below the critical angle). The scattered patterns were obtained at a sample–detector distance of 120 mm using an EIGER2 1M detector (Dectris). The sample distance was calibrated using a NIST LaB$_6$ reference material. The entire beam path was evacuated

to <0.1 mbar. The frames were corrected and integrated in the XSACT software suite[53]. Errors are present owing to the overall measurement scatter, primarily coming from the number of counts per pixel.

## SEM measurements

All SEM measurements were performed on a Cryo-FIB/SEM Aquilos 2 (Thermo Fisher) in OptiTilt mode. The samples were cooled with liquid nitrogen and sputtered with platinum at 1 kV and 30 mA for 15 s to avoid charging effects on the surface. All scans were carried out with an accelerating voltage of 2 kV, a current of 25 pA and a pixel size of around 56 nm. A single frame was acquired with 20 times line integration and a dwell time of 100 ns.

## AFM measurements

All AFM measurements were performed on a NX20 instrument (Park Systems) with non-contact cantilever probes (NSC14/Cr-Au, Park Systems) with a spring constant of 5 Nm$^{-1}$. AFM measurements were conducted in non-contact mode at a scanning rate of 0.4 Hz, an image size of 8 × 8 µm squared and 256 × 256 pixels with image flattening. An initial scan at the sample centre was performed, followed by ten more scans equidistant from the sample centre. For each measurement, the radial PSD was calculated by computing the 2D Fourier transform of the surface roughness data, squaring the absolute values and then averaging over concentric circles to obtain the radial PSD. Averaging over the different regions scanned for each condition (pristine/contacted), we determined the PSD means and error bands. The data shown in Fig. 4h are for the same sample. This was achieved by first performing the scans on the pristine state, then moving the sample out for the 200 contacts with a second linear actuator/force sensor installed in the AFM enclosure (that is, replica of the ones described in Fig. 1b) and then returning the sample under the atomic force microscope for the scans on the contacted state.

## Data availability

The data that support the plots in this paper and other findings of this study are available from the corresponding author on request.

## Code availability

The scripts used to analyse the data in this study are available from the corresponding author on request.

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

**Acknowledgements** This project has received financing from the European Research Council grant agreement no. 949120 under the European Union's Horizon 2020 research and innovation programme. The Analytical Instrumentation Center of the TU Wien acknowledges support by the FFG project 'ELSA' under grant no. 884672. C.M.P. and M.O. acknowledge the state of Lower Austria and the European Regional Development Fund under grant no. WST3-F-542638/004-2021. This research was supported by the Scientific Service Units of the Institute of Science and Technology Austria through resources provided by the Miba Machine Shop, Nanofabrication Facility, Scientific Computing facility, Electron Microscopy Facility and Lab Support Facility. We thank J. Garcia-Suarez and G. Anciaux for the suggestion to look into the roughness power spectral density. We thank I.-M. Strugaru for help with testing the device for Young's modulus measurements.

**Author contributions** J.C.S. performed all contact electrification experiments, analysed the contact electrification data, developed the contact-bias model, performed simulations and wrote the paper. F.P. performed the AFM and Raman experiments, contributed to performing the GIXS, XPS, SEM and LEIS experiments, analysed the corresponding data and wrote the paper. D.M.B. performed the GIXS experiments and analysed the corresponding data. T.C. performed the SEM measurements and analysed the corresponding data. M.S. and A.F. performed the XPS experiments and analysed the corresponding data. M.O. and C.M.P. performed the LEIS data and analysed the corresponding data. Y.W., Y.N. and M.B. performed the HD-SFG experiments and analysed the corresponding data. S.W. conceived the project, secured the funding, contributed to the development of the contact-bias model and wrote the paper.

**Funding** Open access funding provided by Institute of Science and Technology (IST Austria).

**Competing interests** The authors declare no competing interests.

**Additional information**
**Correspondence and requests for materials** should be addressed to Juan Carlos Sobarzo or Scott Waitukaitis.

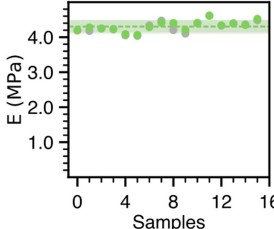

**Extended Data Fig. 1 | Young's modulus before and after 200 contacts.**
We prepare 16 pristine samples and measure their Young's modulus with the
procedure described in the Supplementary Information. We then pair the
samples and perform 200 contacts for each pair, after which we measure their
individual Young's modulus once again.

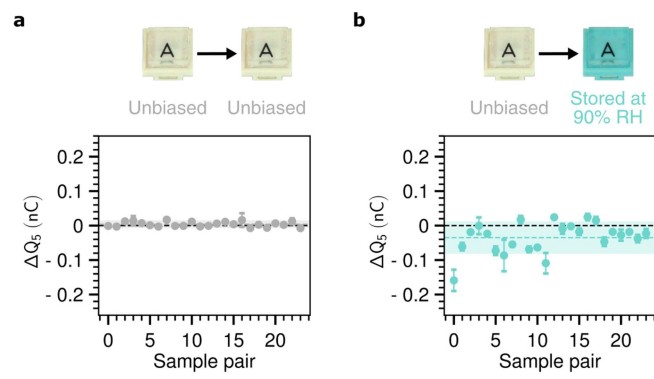

**Extended Data Fig. 2 | Humidity bias. a**, We reproduce the same data presented in Fig. 2a to use as a baseline. **b**, We measure the charge exchange between samples that have been stored at high humidity (roughly 90% RH) and unbiased samples (stored in the main chamber at 30% RH). Samples stored in high humidity charge slightly negatively against unbiased ones.

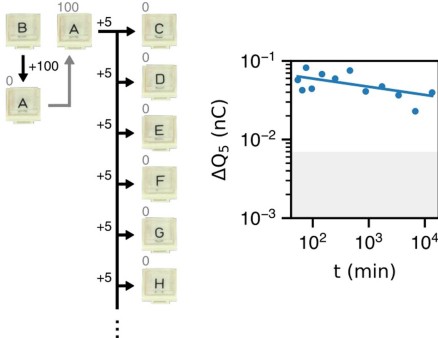

**Extended Data Fig. 3 | Time evolution of the contact bias.** We prepare pristine samples and start by biasing sample A against B with 100 contacts and then proceed to measure the charge exchange between A and samples C–N in certain time intervals. We observe a slight decay in the charge exchange over a scale of days, although it is still much larger than the charge exchange for pristine samples (grey area).

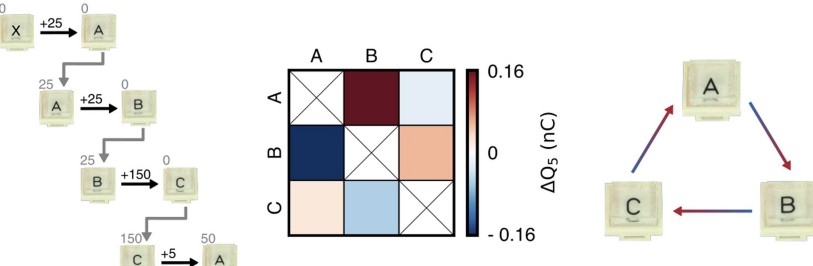

**Extended Data Fig. 4 | Forcing a cycle.** We produce pristine samples A, B, C and X. We start by biasing sample A against X with 25 contacts, then we bias sample B against A with 25 contacts while measuring $\Delta Q_5$. Next, we bias sample C against B with 150 contacts while measuring $\Delta Q_5$. Finally, we measure $\Delta Q_5$ between C and A. The results in matrix form show the appearance of a cycle.

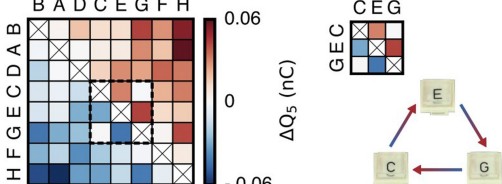

**Extended Data Fig. 5 | Inability to force a series with staggered contact sequence.** We repeat series experiments with the same samples used to force the series of Fig. 3d. The result does not resemble the alphabetical order previously obtained and is not even a perfect series. Furthermore, we observe a cycle, which is a signature of a series in the process of evolution.

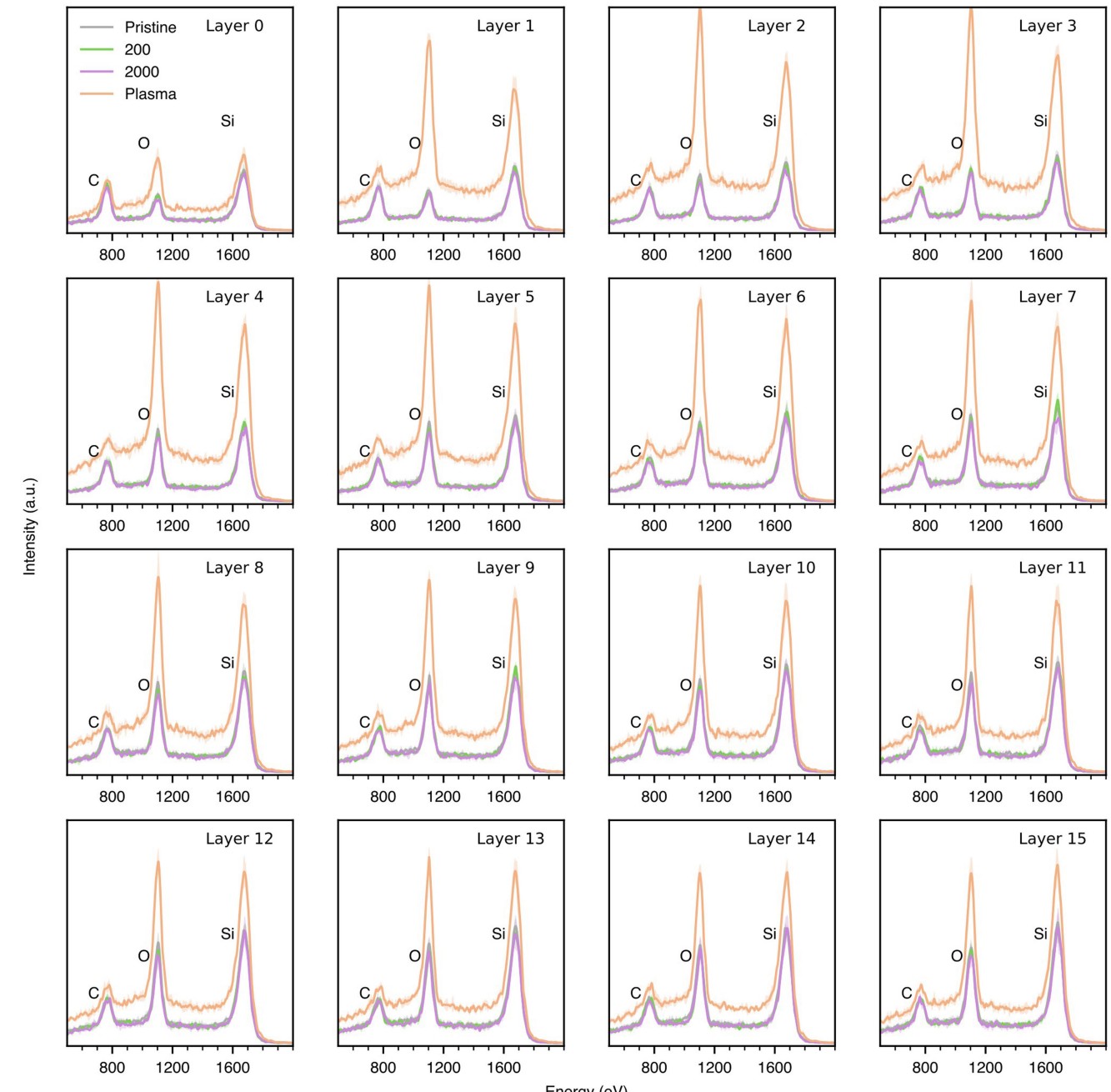

**Extended Data Fig. 6 | Single-layer profiles of LEIS.** Sequential probing and sputtering of the surface reveal no differences between pristine and contacted samples several atomic layers into the bulk (layer 0 being the outermost and layer 15 the innermost). By contrast, the plasma-treated sample shows substantial differences, notably an increased oxygen peak owing to the introduction of OH groups. Furthermore, the deeper layers of the plasma-treated sample approach those of the pristine sample, indicating that this change occurs primarily in the first few layers.

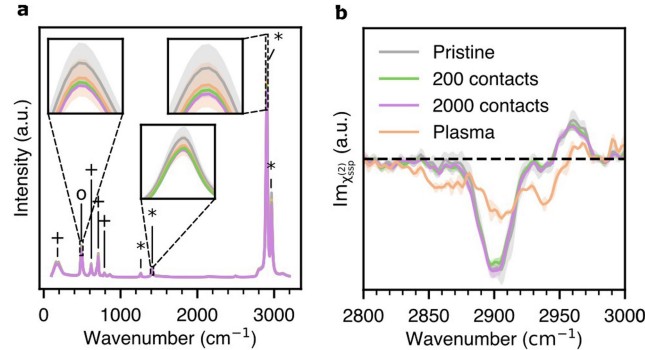

**Extended Data Fig. 7 | Further Raman and HD-SFG measurements. a**, Raman spectroscopy reveals no molecular changes within all samples. The insets highlight an Si–O mode and two C–H modes from left to right, respectively. **b**, For HD-SFG, no differences are detected in the symmetric and asymmetric peaks between pristine and contacted samples. However, the plasma-treated sample clearly shows a much lower signal for the symmetric and asymmetric modes at 2,900 cm⁻¹ and 2,960 cm⁻¹, respectively.

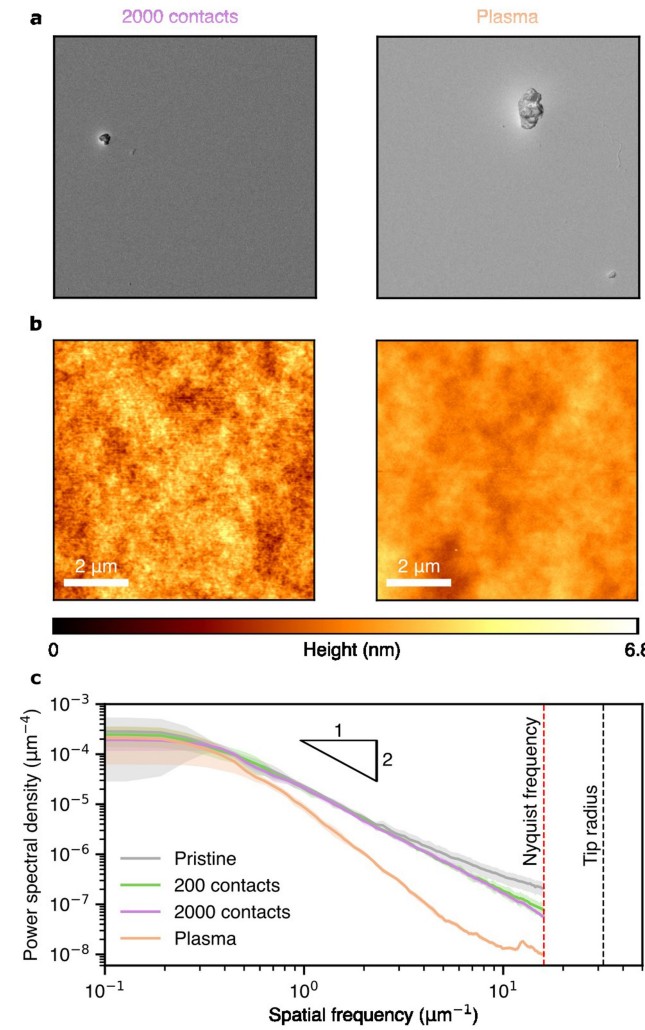

**Extended Data Fig. 8 | Further SEM and AFM scans. a**, We find no visible changes in the surface integrity between SEM scans of the 2,000-contact sample (left) and the plasma-treated sample (right); regions with specks are shown intentionally to aid visualization. **b**, With AFM, the 2,000-contact sample appears visually indistinguishable from 200-contacted or pristine samples. However, plasma treatment smoothes the roughness to an extent that is visibly apparent. **c**, We compare the radial PSD for pristine, 200-contact, 2,000-contact and plasma-treated samples. More contacts progressively smooth the spatially high-frequency features and the plasma treatment changes the roughness across all length scales.

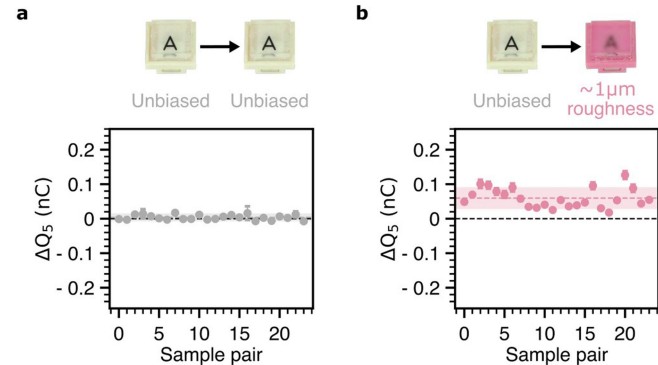

**a** Unbiased → Unbiased

**b** Unbiased → ~1μm roughness

**Extended Data Fig. 9 | Roughness bias. a**, We replot the data presented in Fig. 2a as a baseline. **b**, We measure the charge exchange between samples produced with a high roughness (about 1 μm) and unbiased samples (about 1 nm). High-roughness samples tend to charge positive against low-roughness samples and with a larger magnitude than unbiased against unbiased samples.