## [Peer Review file · Nature]

Spontaneous Ordering of Identical Materials into a Triboelectric Series

Corresponding Author: Mr Juan Carlos Sobarzo

Version 0:

Reviewer comments:

Referee #1

(Remarks to the Author)

The manuscript reports the self-organisation of identical PDMS samples into triboelectric series after contact-separation tests. They have found interesting memory effects – contacted samples will always charge negatively when contacted with fresh ones. Self-ordering in triboelectric series and memory effect has not been reported before. Thus, it may attract broad scientific interest. The manuscript is clearly written, and the data are convincing. They repeated the tests multiple times to be confident about the results represented by small error bars. The provided conclusions are robust.

The reason behind the memory effect and ordering in triboelectric series is correctly identified – the surface smoothening. A decrease in roughness has been measured for the contacted samples. This effect itself has been reported before (ref. 29), where it was demonstrated that the smoother samples on the identical polymer contact always charge negatively.

Unfortunately, the manuscript does not provide a reason behind this phenomenon. Another significant drawback is that the self-ordering in triboelectric series and memory effect is demonstrated only for a single polymer – PDMS. It is not known if this is a general trend for all polymers. Thus, the significance may not be high enough for the Nature.

Referee #2

(Remarks to the Author)

This is a really interesting paper that demonstrates a surprising and hidden rationality in something that seemed like irreproducibility. I think this paper will have a significant impact on the field.

Contact electrification is the effect that you see when rubbing a balloon on your hair, and it's what leads to the shock you get when you touch a doorknob after walking across a rug. Contact electrification is ubiquitous, and without it we wouldn't exist... it likely played a key role in the formation of planets via agglomeration of charged dust particles, and the volcanic lightning caused by charged ash particles likely provided the activation energy for volcanic gases to react to form the biomolecules that gave rise to life. Nowadays, contact electrification underlies entire industries, such as digital printing, but when not controlled it causes problems ranging from fried electronic components to explosions.

While contact electrification seems like science for elementary school children, it is a deceptively complex phenomenon and we have essentially zero understanding of the underlying physics. For example, we don't know what carries the charge (eg, electrons or ions) and there is no way to theoretically predict the direction of charge transfer (ie, which surface charges positive and which charges negative).

One of the frustrating aspects of contact electrification is the low level of reproducibility of charging experiments. Even the direction of charge transfer varies, and different investigations often report different directions of charge transfer for the same pair of materials. A laundry list of reasons has been given for this, but it's all been conjecture.

This is where the present manuscript fits in. The authors show that surface roughness affects the driving force for the direction of charge transfer. While this has been hypothesized before (but never so well demonstrated), the really interesting idea here is that the very act of contact inherent in contact electrification changes the nanoscale surface roughness, thereby changing the driving force governing the direction of charging. Through a series of clever and very-well-controlled experiments, they show that this effect leads to a *predictable* variation in the direction of charge transfer. A simple

minimalistic model is developed that captures what would previously have been considered "unexplained variability". As a smoking gun, the authors show that they can exploit this effect to create any arbitrarily-chosen ordering of samples in regard to the direction of charge transfer, and in this way have turned "unexplained variability" into "rational design".

I recommend that the paper be published after addressing the following points:

1. Line 252: "CE in our system can be definitively connected to some mechanochemical mechanism"

I disagree with this statement. I believe the authors definitively show that the "mechanochemical" changes that occur during contact affect the CE that occurs with subsequent contacts. But I don't think this shows that the contact electrification itself is due to a mechanochemical mechanism. For example, the contact electrification in a single contact can be due to electron transfer, with the ϕ for electron transfer affected by the changes in surface morphology arising from the previous contacts. And similarly, the contact electrification in a single contact can be due to ion transfer, with the ϕ for ion transfer affected by the changes in surface morphology arising from the previous contacts.

2. Line 29: "...supporting a mechanochemical mechanism"

This is ambiguous in regard to what this mechanism is referring to (whether CE or the history-dependence). See above if for CE.

3. I don't fully understand the sorting procedure in Fig 1. I.e, I understand how it is done if there are no conflicts, but how is it done when there are conflicts?. This should be better explained.

4. Does ϕ_{∞} depend on contact conditions, eg contact force? Is it different for shearing contact?

5. minor things:

a. Fig 2... the "d" label is mistakenly a "b"

b. Two terms used for the same idea, should choose one term and go with it. "chemical" vs "molecular" for the same idea (lines 206, 214)

I choose to sign the review report.

Daniel Lacks

Referee #3

(Remarks to the Author)

This paper presents the possibility of the difference of the intrinsic surface charging behavior resulted from the contact and detachment between same materials, and their self-organization in the triboelectric series. I agree that the triboelectric series is empirically verified, which leads the series to possess randomness without agreement into one. However, I think it is hard to consider how the provided strategy contributes to increasing the accuracy of the triboelectric series. In this context, I think it is better to be rejected even if the aim of this manuscript is fit well with Nature.

1) First of all, I understood the organization of the charge induction behavior, but I'm curious about how this organization can help to enhance the reliability of the triboelectric series. In this context, I think author should provide the organization behavior under contact and detachment between different materials, and show the "perfect triboelectric series" with several materials.

2) Author should provide scientific reason about the different charge induction behavior after contact and detachment between same materials, due to the general knowledge about the contact electrification, which is the surface charging phenomenon under the contact and detachment between different materials.

3) I think author should add detailed explanation about the experiments. For example, to explain the experiments in Figure 2C and 2D, I guess author should explain about reason of using "lagging sample", "advancing sample" and "extra sample" for the experiment, and the reason of experimental setup in Figure 2C. The lack information about the experiments hinders the clear understanding of the whole manuscript. Also, I think author should provide reason about the experimental model in Figure 3c.

4) I'm curious about the "self-organization process". Does the author use the AI models to arrange the experimental data? How the data can be self-organized?

5) I think author should provide the necessity of the analysis demonstrated in Figure 4. Right now, I cannot understand the relationship between the self-organization of the triboelectric series and Figure 4.

Referee response for

'Spontaneous Ordering of Identical Materials into a Triboelectric Series'

We thank the referees for their careful reading of our manuscript and their constructive comments. Below we respond to their reports. Their comments are in *italicized Times New Roman font*, our responses are in **blue sans serif**.

First Referee

The manuscript reports the self-organisation of identical PDMS samples into triboelectric series after contact-separation tests. They have found interesting memory effects – contacted samples will always charge negatively when contacted with fresh ones. Self-ordering in triboelectric series and memory effect has not been reported before. Thus, it may attract broad scientific interest. The manuscript is clearly written, and the data are convincing. They repeated the tests multiple times to be confident about the results represented by small error bars. The provided conclusions are robust.

We thank the referee for accurately summarizing our findings and their potential for broad impact.

The reason behind the memory effect and ordering in triboelectric series is correctly identified – the surface smoothening. A decrease in roughness has been measured for the contacted samples. This effect itself has been reported before (ref. 29), where it was demonstrated that the smoother samples on the identical polymer contact always charge negatively. Unfortunately, the manuscript does not provide a reason behind this phenomenon.

In our opinion, why roughness matters in same-material CE is still a big open question. We wouldn't claim to resolve this question without rock-solid experimental evidence—in particular at the nanoscale and smaller—which we don't have. Nonetheless, at least two compelling hypotheses exist in the literature, which we discuss more fully here and in the improved closing paragraph of the new manuscript.

For instance, Verners [29] *et al.* proposed the following three-part model to explain why roughness matters. First, contact of nanoscale protrusions creates large strains. Second, these can cause heterolytic bond scission to release charged species for transfer. Third, the negative charged fragments generally have smaller desorption energies than positive ones, hence leading to a preferential sign for transfer. Beyond a few qualitative macroscopic observations, Verners' evidence for this hypothesis is essentially theoretical in the form of FEM and RMD simulations.

An alternative hypothesis comes from Marks and colleagues [24], who propose that roughness comes in via coupling to flexoelectricity. In this idea, the strain gradients of nanoscale asperities are what matter, as they lead to large electric fields during contact that can drive free charge. Again, this is very interesting, but so far it is still essentially a theoretical idea, as the experimental support is largely macroscopic and qualitative.

We would love to be in a position to experimentally validate/invalidate either of these hypotheses, but this is no small task. To validate the Verners' idea, one must: (a) measure the release of ionic fragments during contact, (b) measure their 'desorption energies', and (c) develop a quantitative model accounting for *spectral* roughness, not merely the RMS roughness. To validate the Marks' idea, one must: (a) measure the flexoelectric coefficient(s), (b) experimentally identify the source of free charge, and (c) again develop a detailed model accounting for the spectral roughness. In our opinion, overcoming these challenges and identifying what's really going on at the smallest scales is not going to be achieved in any single paper, but rather by a large-scale, long-term effort of the entire community.

In the new manuscript, we give more thorough attention to the ideas of Refs. [29] and [24]. In our opinion, they are the leading ideas, but there's simply too big a disconnect between their microscopic underpinnings and macroscopic experiments to decide which, if either, occurs. In a departure from the tone of our previous manuscript, we allow ourselves to speculate a bit, suggesting there is no reason that aspects of *both* models could be at play. We of course put a clear asterisk next to this speculation.

Another significant drawback is that the self-ordering in triboelectric series and memory effect is demonstrated only for a single polymer – PDMS. It is not known if this is a general trend for all polymers. Thus, the significance may not be high enough for the Nature.

So far we have limited our studies to PDMS, but this is primarily due to practicalities. As we explain in the manuscript, PDMS is well-suited to our work because we can easily fabricate it in house and the resulting samples are extremely soft and smooth. These qualities ensure reproducible and 'conformal' contacts. Additionally, our apparatus is limited in the maximum force that can be applied, preventing us from using most other polymers considering how stiff they are (~1000X stiffer than PDMS).

Yet there is good reason to imagine that the effect we observe is widespread. As the Referee can probably appreciate, Verners' work shows that the effect of 'smooth polymers charging negatively' is widespread [29]. If those results are taken at face value, the only conceptual leap required to imagine the prevalence of spontaneous ordering is the prevalence of surface alterations during contact. Yet this notion is foundational in tribology, forming the basis for Bowden and Tabor's theory of friction and wear [45].

Exploring the effect with many polymers and parameters (e.g. contact force, contact duration, number of contacts, bias force, biasing against other materials, biasing via roughening instead of smoothing...) comprises a huge phase space that we're not going to be able to cover in one paper. However, we think it's going to be a really rich direction for future work throughout the community, and should keep our friends and colleagues busy for some time now that the notion of 'contact history' will be in the CE lexicon.

Second Referee

This is a really interesting paper that demonstrates a surprising and hidden rationality in something that seemed like irreproducibility. I think this paper will have a significant impact on the field.

Contact electrification is the effect that you see when rubbing a balloon on your hair, and it's what leads to the shock you get when you touch a doorknob after walking across a rug. Contact electrification is ubiquitous, and without it we wouldn't exist... it likely played a key role in the formation of planets via agglomeration of charged dust particles, and the volcanic lightning caused by charged ash particles likely provided the activation energy for volcanic gases to react to form the biomolecules that gave rise to life. Nowadays, contact electrification underlies entire industries, such as digital printing, but when not controlled it causes problems ranging from fried electronic components to explosions.

While contact electrification seems like science for elementary school children, it is a deceptively complex phenomenon and we have essentially zero understanding of the underlying physics. For example, we don't know what carries the charge (eg, electrons or ions) and there is no way to theoretically predict the direction of charge transfer (ie, which surface charges positive and which charges negative).

One of the frustrating aspects of contact electrification is the low level of reproducibility of charging experiments. Even the direction of charge transfer varies, and different investigations often report different directions of charge transfer for the same pair of materials. A laundry list of reasons has been given for this, but it's all been conjecture.

*This is where the present manuscript fits in. The authors show that surface roughness affects the driving force for the direction of charge transfer. While this has been hypothesized before (but never so well demonstrated), the really interesting idea here is that the very act of contact inherent in contact electrification changes the nanoscale surface roughness, thereby changing the driving force governing the direction of charging. Through a series of clever and very-well-controlled experiments, they show that this effect leads to a *predictable* variation in the direction of charge transfer. A simple minimalistic model is developed that captures what would previously have been considered "unexplained variability". As a smoking gun, the authors show that they can exploit this effect to create any arbitrarily-chosen ordering of samples in regard to the direction of charge transfer, and in this way have turned "unexplained variability" into "rational design".*

We thank the referee for the comprehensive summary of the state of the field and the potential impact of our findings, as well as the supportive comments.

I recommend that the paper be published after addressing the following points:

1. Line 252: "CE in our system can be definitively connected to some mechanochemical mechanism"

I disagree with this statement. I believe the authors definitively show that the "mechanochemical" changes that occur during contact affect the CE that occurs with subsequent contacts. But I don't think this shows that the contact electrification itself is due to a mechanochemical mechanism. For example, the contact electrification in a single contact can be due to electron transfer, with the ϕ for electron transfer affected by the changes in surface morphology arising from the previous contacts. And similarly, the contact electrification in a single contact can be due to ion transfer, with the ϕ for ion transfer affected by the changes in surface morphology arising from the previous contacts.

We agree with the referee. We have changed this statement for a more thorough discussion that conveys the implications of our results appropriately.

2. Line 29: "...supporting a mechanochemical mechanism"

This is ambiguous in regard to what this mechanism is referring to (whether CE or the history-dependence). See above if for CE.

Similarly, we have changed this line for a more accurate statement.

3. I don't fully understand the sorting procedure in Fig 1. I.e, I understand how it is done if there are no conflicts, but how is it done when there are conflicts?. This should be better explained.

We agree with the referee, and consequently we have included a more thorough explanation of the matrix sorting process in Methods.

4. Does ϕ_{∞} depend on contact conditions, eg contact force? Is it different for shearing contact?

We performed additional experiments to test this idea, analogous to those of Fig. 2d. Such experiments quickly open up a large phase space of questions/parameters: What is the 'measurement force'? What is the 'bias force'? At how many contacts is the switch made?

To scratch the surface of this large phase space, we started out by using our usual contact force of ~ 4.5 N ($\sim 1\%$ strain) for all measurements of ΔQ_5 . For biasing (i.e. contacts between the advancing sample and extra sample), we used this same force *until* the advancing sample reached 420 contacts (at which point the lagging sample had 20), that is, the advancing sample had been biased four times. Then we changed the force for biasing to ~ 9 N ($\sim 2\%$ strain). We remark that we kept the 'measurement' force of ~ 4.5 N (i.e. between the advancing and lagging samples). With this particular initial test, we saw no significant changes on the behavior of ϕ^{∞} .

To do a full search of phase space, we would need to: (a) upgrade our apparatus to work at much larger and smaller forces (we are currently limited to ~ 10 N), and (b) upgrade our apparatus to be much more automated than it already is (e.g. switching samples automatically and discharging *in situ*). Such experiments promise to be very interesting, but also quite hard without these improvements.

Unfortunately, we are not able to perform shear contact with our setup. Building this idea into a 'rheometric device' would be interesting, but beyond what we can currently do.

5. *minor things:*

a. *Fig 2... the "d" label is mistakenly a "b"*

b. *Two terms used for the same idea, should choose one term and go with it. "chemical" vs "molecular" for the same idea (lines 206, 214)*

We appreciate the corrections and we have updated the text and figure accordingly.

I choose to sign the review report.

Daniel Lacks

Third Referee

This paper presents the possibility of the difference of the intrinsic surface charging behavior resulted from the contact and detachment between same materials, and their self-organization in the triboelectric series. I agree that the triboelectric series is empirically verified, which leads the series to possess randomness without agreement into one. However, I think it is hard to consider how the provided strategy contributes to increasing the accuracy of the triboelectric series. In this context, I think it is better to be rejected even if the aim of this manuscript is fit well with Nature.

The intent of our paper is not to 'increase the accuracy of the triboelectric series'. Rather, the intent is to show that the *contact history* of samples can alter the way they exchange charge. To our knowledge, this idea has never been considered in the CE literature, let alone experimentally measured and explained. The primary value of our work is to bring this unanticipated and impactful idea to light. The secondary value is the consequences it has for CE more broadly, in particular explaining why CE is so unpredictable. We hope our data encourages the community to consider the *contact history* of samples when performing experiments, which may lead to more 'accurate' results on many levels.

1) First of all, I understood the organization of the charge induction behavior, but I'm curious about how this organization can help to enhance the reliability of the triboelectric series. In this context, I think author should provide the organization behavior under contact and detachment between different materials, and show the "perfect triboelectric series" with several materials.

Similar to what we mentioned before, our intent is not to 'enhance the reliability of the triboelectric series'. Rather, our intent is to show the Referee something they have probably never considered or seen: that identical materials spontaneously order into TE series and that this ordering is caused by contact itself.

Regarding other materials, we do expect that a variety of effects could be seen. For instance, we think the positions in TE series of different materials could potentially switch due to contact biases. Exhibiting this would be very interesting, but is outside of the claims and scope of our current paper.

2) Author should provide scientific reason about the different charge induction behavior after contact and detachment between same materials, due to the general knowledge about the contact electrification, which is the surface charging phenomenon under the contact and detachment between different materials.

The mechanism and carriers in CE are not agreed upon. Hence, we can't explain our results in the context of general knowledge because there is very little agreed upon knowledge. There are probably half a dozen 'serious' proposals in the literature for how CE occurs between different insulators, but none of these have dominant support in the community. See for example the review article by Lacks and Shinbrot [1].

A big part of the value of our work is that it distills parameters relevant to CE's unknown mechanism. For instance, our data shows quite cleanly that roughness alone can control the sign of charge between two nominally identical samples. Henceforth, models that can't account for this should be considered with skepticism.

3) I think author should add detailed explanation about the experiments. For example, to explain the experiments in Figure 2C and 2D, I guess author should explain about reason of using "lagging sample", "advancing sample" and "extra sample" for the experiment, and the reason of experimental setup in Figure 2C. The lack information about the experiments hinders the clear understanding of the whole manuscript. Also, I think author should provide reason about the experimental model in Figure 3c.

We updated the explanation of the experiments in the Methods.

The samples are named like this because we are creating a growing difference in their contact history. The 'advancing' sample is named so because its number of contacts 'advances' faster than the 'lagging' sample, whose contacts 'lag' behind.

In Fig. 2a,b we show that contact itself creates a preferred direction of charging—previously contacted samples charge negatively against uncontacted samples. The motivation for the experiments illustrated in Fig. 2c,d is to observe the evolution of this 'contact bias', specifically how it behaves when the 'contact difference' between the advancing/lagging samples becomes larger.

As we describe in the manuscript, Fig. 3c illustrates the algorithm we use to 'force' a series of our choosing, in this case alphabetical order of the samples from A to H.

4) I'm curious about the "self-organization process". Does the author use the AI models to arrange the experimental data? How the data can be self-organized?

The self-organization does not refer at all to how we organize the data. It refers to how a set of identical samples that initially charge randomly (*i.e.* they do not order into a series), can change in such a way that they eventually *do* order into a series.

In the Methods of the updated manuscript, we explain in more detail how we sort the charging matrices, which will help answer the referee's question. We do not make use of any AI models for any purpose.

5) I think author should provide the necessity of the analysis demonstrated in Figure 4. Right now, I cannot understand the relationship between the self-organization of the triboelectric series and Figure 4.

Figures 1-3 show (i) that samples self-organize into a triboelectric series (Fig. 1), (ii) that this is driven by the act of contact (Fig. 2), (iii) that this can be explained and controlled by a contact-history-based framework (Fig. 3). The natural follow-up question to Figs. 1-3 is: what are contacts actually *changing* in the surfaces of our samples? The purpose of Fig. 4 is to address this, which we do via battery of surface-sensitive tests. XPS and LEIS data show that there are no measurable changes to the atomic composition in the surface. Raman and HD-SFG data show that there are no changes to the molecular composition of the surface. In the end, the only observable changes are in the spectral properties of roughness. Hence, one can conclude with some degree of confidence that the spontaneous ordering of the triboelectric series is due to these changes in nanoscale roughness.